# Two-Dimensional Amphibian Diversity along a 3500 m Elevational Gradient at the Eastern Edge of the Qinghai–Xizang Plateau

**DOI:** 10.3390/ani14121723

**Published:** 2024-06-07

**Authors:** Jiaxing Tang, Jiaxin Cui, Gang Wang, Yong Jiang, Huaming Zhou, Jianping Jiang, Feng Xie, Jie Wang, Guiying Chen

**Affiliations:** 1College of Life Sciences, Sichuan Normal University, Chengdu 610101, China; jiaxingt@foxmail.com; 2CAS Key Laboratory of Mountain Ecological Restoration and Bioresource Utilization & Ecological Restoration and Biodiversity Conservation Key Laboratory of Sichuan Province, Chengdu Institute of Biology, Chinese Academy of Sciences, Chengdu 610213, China; cuijiaxin23@mails.ucas.ac.cn (J.C.);; 3College of Chemistry and Life Sciences, Chengdu Normal University, Chengdu 611130, China; wanggang8793@163.com; 4Sichuan Gongga Mountains National Nature Reserve, Kangding 626000, China; 5Ganzi Tibetan Autonomous Prefecture Forestry Research Institute, Kangding 626001, China

**Keywords:** distribution pattern, Gongga Mountains, species communities, species richness, elevation range

## Abstract

**Simple Summary:**

Gonga Mountains stretch latitudinally and locate at the eastern edge of the Qinghai–Xizang Plateau and the center of Hengduan Mountain Range, a global biodiversity hotspot. In this study, 20 amphibian species from seven families and twelve genera were identified in the Gongga Mountains through field investigation and museum specimen retrieval. For the horizontal dimension, the amphibian species composition in the western slope was significantly different from that in the eastern slope. For the vertical dimension, the amphibian species richness displayed a unimodal pattern, peaking at mid elevation, 3300–3700 vs. 1700–1900 m a.s.l. in the western and eastern slopes, respectively. The maximum species diversity occurred in different types of vegetation, i.e., alpine coniferous forests on the western slope, and evergreen broad-leaf forests and coniferous broad-leaf forests on the eastern slope. This discovery highlights the complex biodiversity in the Hengduan, Mountains and underscores the need for targeted management and protection.

**Abstract:**

Amphibians serve as reliable indicators of ecosystem health and are the most threatened group of vertebrates. Studies on their spatial distribution pattern and threats are crucial to formulate conservation strategies. Gongga Mountains, with a peak at 7509 m a.s.l. and running latitudinally, are in the center of the Hengduan Mountains Range and at the eastern steep edge of the Qinghai–Xizang Plateau, providing heterogeneous habitats and varied niches for amphibians. In this study, we combined 83 days of field work with information from 3894 museum specimens that were collected over the past 80 years, and identified twenty amphibian species belonging to seven families and twelve genera by morphology. Of these species, seven were listed in the threatened categories of the Red List of China’s Biodiversity and thirteen were endemic to China. Ten species were found on the plateau side (western slope) and eleven species were found on the other side close to the Sichuan Basin (eastern slope). Only one species was found on both sides, indicating different community structures horizontally. The species richness was unimodal vertically and peaking at mid elevation on both sides, with the maximum number (ten vs. nine) of species occurring at 3300–3700 vs. 1700–1900 m a.s.l. and in different types of vegetation. The elevation span and body length of species distributed on both slopes did not show significant differences. These findings help to understand the horizontal and vertical distribution pattern of amphibian diversity, laying a foundation for future biogeographical and conservation research in this area.

## 1. Introduction

Biodiversity is distributed heterogeneously across the earth, discerning its pattern across environmental gradients, and causal mechanism has long been a core objective in the fields of biogeography and ecology [1]. A well-defined macroecological pattern is the decrease in taxa-specific biodiversity with elevation [2]. There are four predominant patterns commonly observed in species richness along elevational gradient: decreasing, low plateau, low plateau with a mid-elevational peak and mid-elevation peak [3,4]. Mountainous regions offer a significant opportunity to explore the pattern, where rapid changes in temperature and precipitation generally occur over short distances [5,6]. This may lead to a high biodiversity, often with sharp transitions in vegetation sequences in mountain areas [7,8]. In addition, mountainous regions exhibit higher levels of endemism among species [8]. However, with a growing understanding of ongoing climate change, it is increasingly clear that mountains are particularly vulnerable to the impacts of climate change due to the amplification of warming effects at higher elevations [9,10].

Compared to other vertebrate groups, amphibians are known for their particular sensitivity to environmental changes [11], and are both more threatened and declining more rapidly than birds or mammals [12,13]. Serving as predators and prey, amphibians are an important part of the ecosystem, playing a vital role in maintaining the stability of the food web [14], and are reliable indicators of environmental stress due to their special ecological needs [15,16]. Amphibians generally inhabit specific environments due to the physiological demands of water balance and thermoregulation [17]. There is evidence that smaller geographic ranges are associated with extinction risk for amphibians [18,19].

Forests have most of the terrestrial biodiversity on earth [20] and provide habitats for 70% of the world’s amphibians [21]. The rich microhabitats of forest floors provide a suitable living environment for amphibians, playing an irreplaceable role in their foraging, reproduction, and wintering [22,23]. The reconstruction of vegetation and forest cover helps to promote the distribution of amphibians, and provides shelter and resources for their resettlement, feeding and breeding [21,22]. Furthermore, as ectotherms, amphibians are expected to show a significant response to thermal changes along an altitude gradient [24]. By studying the ecological niche of amphibians, we can better understand the structure and function of the ecosystem, and provide scientific support for their protection and restoration, which is of great significance for maintaining the stability and anti-interference ability of the ecosystem [25].

The Gongga Mountains, including a peak of 7509 m, are located at the eastern edge of the Qinghai–Xizang Plateau. It is also the highest peak of the Hengduan Mountains, which is one of the world’s 36 biodiversity hotspots [26]. The Qinghai–Xizang Plateau has a unique environment characterized by low temperature, low humidity and low air pressure [27]. In contrast to the deforested area that border the Sichuan Basin, the Gongga Mountains are still covered with primary forests in many places [28]. However, the ecosystem in the Gongga Mountains faces many threats and challenges. With the continuous expansion of human activities, deforestation, land reclamation and road construction are rapidly destroying the original ecological environment in this area. In addition, climate change has also led to changes in temperature and precipitation patterns, affecting amphibian habitats and the availability of food resources. Under the combined pressure of these factors, amphibians in the Gongga Mountains may face severe challenges to survive.

The objective of this study is to compile and update data on the species richness and threat status of amphibians in the Gongga Mountains. Additionally, we aim to analyze the distribution pattern of these species using a combination of field surveys, literature reviews, and the retrieval of museum specimens. This research will contribute to the understanding of the vertical and horizontal distribution of amphibian diversity, providing a basis for future biogeographical analysis in this region. Furthermore, the findings will assist with the establishment of conservation priorities and targets.

## 2. Materials and Methods

### 2.1. Study Area

The Gongga Mountains run from north to south, at the intersection of the eastern edge of the Qinghai–Xizang Plateau and the western side of the Sichuan Basin (Figure 1A), resulting in an elevational range exceeding 6400 m (1100–7509 m a.s.l.) within a span of less than 30 km [29]. According to the direction of the southeast monsoon, the Gongga Mountains are separated into the windward (eastern) and leeward (western) slopes [30]. The annual precipitation varies from 1068 to 3210 mm, while the mean annual temperature ranges between 22.6 and 14.5 °C, depending on the elevation. Under the influence of the southwest monsoon and extra-tropical westerlies, the climate on the western slope is warmer and drier, whereas it is relatively cooler and humid on the eastern slope [28]. At the same elevation, the western slope is 1–3 °C warmer than the eastern slope [30].

The average temperature decreases by 0.6 °C for every 100 m rise in altitude [31]. Due to the large altitude and temperature difference and the influence of the southeast monsoon, the eastern slope has formed complete and diverse vertical vegetation belts. From low to high elevation, there are arid shrubs and grasses (<1200 m), evergreen broad-leaf forests (1200–2200 m), coniferous and broad-leaf forests (2200–2500 m), subalpine coniferous forests (2500–3600 m), alpine scrub meadow (3600–4600 m), alpine screes (4600–4900 m) and permafrost (above 4900 m) [4,28,30,32,33]. The western slope has less precipitation and distinct dry and wet seasons, resulting in simpler vegetation types (Figure 1C,D). The elevation distribution of the vegetation zone is slightly upward compared with that of the eastern slope, and it has only four zones: subalpine coniferous forests (2600–4000 m), alpine scrub meadow (4000–4800 m), alpine screes (4800–5100 m) and permafrost zones (above 5100 m) [4,28,30,32,33].

This study was conducted on both slopes of the Gongga Mountains (101.483–102.200° E, 29.017–30.083° N). We take the ridges and valleys as the demarcation, specifying that the eastern slope of the Gongga Mountains extends to the Dadu River Valley, while the western slope ends at the Liqiu River valley (Figure 1B).

### 2.2. Field Survey and Museum Specimen Retrieval

Field surveys were conducted during nine breeding seasons between 2001 and 2017. One hundred and eighty-three line transects, each with a length of 150–700 m, were set along rivers or streams. Additionally, several alpine lakes and reservoirs were also surveyed. During the survey, two investigators walked along the stream at a relatively steady pace of about 2 km/h, carefully inspecting rocks during the daytime (10:00–12:00 and 15:00–17:00) or nighttime (19:30–23:30) with a flashlight. Encountered animals (adults or juveniles) were captured, immediately identified by morphology referring to a color atlas of Chinese amphibians [34], and then released. For those that were difficult to confidently identify, specimens were collected and preserved for further examination in the lab. Larvae, in their early stages, were challenging to distinguish from each other. To avoid misidentification, larvae were morphologically identified only after the hind legs developed. For each encounter, data on species, number of individuals, GPS position, and microhabitat type were recorded.

Complementary information was collected from specimens stored in the herpetological museum of the Chengdu Institute of Biology, Chinese Academy of Sciences. Most specimens stored there were collected and identified by renowned herpetologists, including Chengchao Liu, Liang Fei, Guanfu Wu, and Ermi Zhao. A review was conducted on the records of 3894 amphibian specimens collected from the Luding, Kangding and Jiulong counties during 1938–2008. We collapsed specimens from the same location and date into a single survey, resulting in 51 field surveys and 472 field sites across 33 years (refer to Appendix A). These historical surveys were also examined in accordance with published references [34,35].

We generated a comprehensive database that includes elevational distribution data (minimum and maximum elevations of occurrence) and snout-to-vent length [36], as well as the threat status of each species. Species nomenclature followed the Amphibian Species of the World v6.2 (https://amphibiansoftheworld.amnh.org/ accessed on 20 January 2024).

### 2.3. Elevational Range Size Analysis

For each species, we plotted the local elevational range (maximum minus minimum elevation) observed in the Gongga Mountains and compared it with the overall elevational range [37] throughout its distribution range, which was retrieved from Amphibian in China (https://www.amphibiachina.org/amphibia accessed on 20 January 2024) and references [34,35], and checked for accuracy. To assess the threat status of each species, we consulted both the IUCN Red List (IUCN 2023) and the Red List of China’s Biodiversity (2021), comparing their threat levels at both global and national scales.

The number of species at different elevations were analyzed by dividing the elevational range into 200 m bandwidths. We counted the number of species at each elevational gradient on the eastern and western slopes.

## 3. Results

### 3.1. Species Composition and Conservation Status

Amphibians were found to distribute across elevations ranging from 1100 to 4600 m within the study area. Twenty amphibian species were morphologically identified according to the 60 field surveys during 1938–2017. These species belong to twelve genera, seven families, and two orders (Table 1, Appendix A). Three species were Caudata and 17 were Anura. The family Megophryidae has the most abundant species (*N* = 7), followed by Ranidae (*N* = 5). Hynobiidae has three species, whereas other families have only one or two species. With reference to the IUCN Red List, six species were categorized as threatened, including one Endangered (EN) (*Scutiger jiulongensis*) and five Vulnerable (VU) species.

Referred to the Red List of China’s Biodiversity, four VU species remain the same category, in contrast, *S. jiulongensis* was downgraded to be listed as VU and *Amolops loloensis* downgraded as Near Threatened (NT), while *Oreolalax major* was upgraded to be listed as VU (Appendix A). Three of the six threatened species belong to Caudata, among which mountain stream salamanders *Batrachuperus karlschmidti* and *B. pinchonii* have very limited elevational ranges and are confined to higher altitudes.

### 3.2. Elevational Range Size of Amphibians in the Gongga Mountains

Aside from Hylidae and Salamandridae (each containing only one species), five families are distributed on both slopes (Table 1). According to our survey, there are 10 amphibian species on the western slope and 11 amphibian species on the eastern slope (Table 1). The species on the western slope are quite different from those on the eastern slope, except that *Bufo gargarizans* are distributed on both slopes. Among the 20 species, only three (*Amolops loloensis*, *A. xinduqiao*, *Nanorana pleskei*) had expanded both upper and lower limits of elevation in the Gongga Mountains in comparison to their overall elevation ranges. However, more than half (11 species) had smaller elevation ranges than their overall elevation ranges. The species with the highest elevation is *Nanorana pleskei*, reaching 4600 m a.s.l. The species with the lowest elevation is *B. gargarizans*, which has the largest local elevation range (1100–3764 m). Fifteen species have an elevation range of less than 1500 m, of which five are listed as VU and three (*Quasipaa boulengeri*, *Oreolalax major* and *Liangshantriton taliangensis*) have extremely narrow elevational ranges (Table 1). Only the cascade frog *Amolops loloensis* and *Bufo gargarizans* have elevation ranges of more than 2000 m, and neither are threatened [38].

### 3.3. Elevational Distribution Pattern

The lowest elevation on the western slope is the valley of Liqiu River (2600 m), much higher than that on the eastern slope (valley of Dadu River, 1100 m). The richness of amphibian species at a certain elevational gradient does not increase with the enlargement of the area (Figure 2B). Due to the significant vertical drop in altitude on the eastern slope, the area of each elevation gradient is relatively similar and small. The peak of species richness occurs within the altitude range of 1500–2500 m, decreasing rapidly with increasing altitude.

Conversely, on the western slope, the high-altitude areas within each elevation gradient have larger areas. The peak of species richness occurs between 3100 and 3900 m. The distribution of amphibians on both eastern and western slopes exhibits a unimodal pattern (Figure 2C). In the western slope, the peak of species richness (10 species at 3300–3700 m) is slightly higher than that in the eastern slope (9 species at 1700–1900 m). Additionally, four high-altitude toad species (*Scutiger* spp.) inhabit the western slope at 3300–4200 m, representing a special group endemic to the high plateau.

Vegetation types on the western slope generally occur 200–400 m higher than those on the eastern slope. Amphibians on the eastern slope primarily inhabit evergreen broad-leaf forests, coniferous and broad-leaf forests, and subalpine coniferous forests, while they mainly inhabit subalpine coniferous forests and alpine scrub meadow on the western slope (Figure 2A). The number of species in the alpine scrub meadow differs greatly between slopes, with seven species on the western slope, but only one species (*Bufo gargarizans*) on the eastern slope.

The elevation range of species on the western slope is higher, but not significantly different between slopes (Figure 2D). Similarly, the snout-to-vent length (SVL) of frog species on both slopes shows no significant difference (Figure 2E).

## 4. Discussion

### 4.1. Species Richness of Amphibians along the Elevational Gradient in the Gongga Mountains

The distribution of amphibians is greatly sensitive to environmental changes [39], and the study of their distribution pattern is significant for habitat protection and environmental monitoring. We recorded twenty amphibian species belonging to seven families and twelve genera during the 60 field surveys in the past 80 years. Amphibian diversity is generally expected to decrease or exhibit a unimodal pattern with the increasing altitude [3,40,41,42], which is consistent with our findings in the Gongga Mountains. Although the elevation gradient with the highest species richness on the two slopes differs, they both exhibit a single-peak pattern, and species are the most abundant at middle elevations. As they are ectotherms, climatic factors influence various life activities of amphibians. With increasing altitude, temperatures generally decrease monotonically, while precipitation typically peaks at mid-altitudes [3,40,41,42]; therefore, in mid-altitude areas, temperature and humidity remain relatively stable, contributing to the survival and reproduction of amphibians. Additionally, mid-altitude regions usually feature numerous streams, rivers and lakes, providing ample habitats for amphibians. Apart from precipitation, vegetation diversity is also a major factor affecting the species richness of amphibians [43]. Mid-altitude areas have rich vegetation types, especially on the eastern slopes, spanning three vegetation zones. The forest system provides food resources and concealed habitats for amphibians [22], promoting their population stability and prosperity. The eastern slope of the Gongga Mountains boasts a richer variety of vegetation, providing amphibians with more options for survival and resulting in greater differences among species to adapt to diverse environments. On the other hand, although the western slope hosts one fewer species of amphibians compared to the eastern slope, the presence of only two vegetation zones, namely subalpine coniferous forest and alpine scrub meadow, and the distribution of all ten amphibian species within the subalpine coniferous forest on the western slope may intensify interspecies competition.

### 4.2. Threat Factors and Conservation Measures

Amphibians face a deteriorating environment due to the development of tourism, the construction of roads and tourism facilities, which occupy a large amount of land and reduce vegetation coverage [44,45,46,47,48]. The western slope of the Gongga Mountains has fewer vegetation types (Figure 2A) [31], and the ecological niches of livestock and wild mammals are highly overlapping, leading to intense competition between them [49]. The fragile ecological environment of the high-altitude western slope places greater niche selection pressure on amphibians. In contrast, the eastern slope has more diverse vegetation types. However, coniferous and broad-leaf forests cover an elevation belt of just 300 m, suggesting that attention should be paid to the restoration of vegetation to protect the eight amphibian species within this particular elevation belt.

Caudata are among the most threatened groups of vertebrates [12,13]. All three Caudata species in the Gongga Mountains are classified as vulnerable and are listed as national second-class protected animals in China. The Taliang crocodile newt *L. taliangensis* is exclusively found in a single subalpine lake situated at an altitude of 2314 m. Human activities have been observed to impact this habitat, including grazing cattle around the lake and releasing invasive salmonids within its waters. The salmonids could directly predate upon amphibian eggs and larvae, and may also transport deadly pathogens [50,51]. A recorded breeding site in Wanba Town may have been destroyed due to hydropower station construction, and no *L. taliangensis* were found during our four surveys, suggesting that it may have disappeared from that location. Vulnerable species such as *Scutiger jiulongensis*, *Batrachuperus karlschmidti* and *B. pinchonii* inhabit relatively high altitudes. These cold-adapted amphibians are particularly susceptible to the threats of climate change and require ongoing monitoring.

The altitude range of specie is an important factor reflecting community structure and environmental niche [52,53]. Different conservation strategies should be adopted for species with different elevation ranges [41,54]. Small-elevation-ranged species tend to be less abundant, less mobile and poorly competitive, and have narrower habitat breadths compared to large-elevation-ranged species [49]. Consequently, a limited altitude range may indicate that the organism is at a higher risk of extinction [37].

The distribution of amphibians on both slopes of the Gongga Mountains exhibits a unimodal trend. Therefore, prior protection strategies should be implemented in areas at 1500–2500 m (eastern slope) and 3100–3900 m (western slope), where amphibian diversity is the highest. Conservation efforts should focus on maintaining the integrity and complexity of ecological environments by reducing grazing pressure and protecting vegetation cover, with a particular emphasis on species with small ranges. Additionally, efforts should be made to enhance biodiversity protection awareness and emphasize the crucial role of amphibians in ecosystem functioning.

## 5. Conclusions

We assessed the species composition, distribution, and conservation status of amphibians on both the eastern and western slopes of the Gongga Mountains. We analyzed the distribution pattern of amphibians on two slopes and discovered that both exhibited unimodal trends, with peaks at middle elevation. There were no significant differences in elevation span or body length between species on the two slopes. However, the distribution of species on the western slope is more concentrated in the high-altitude range, with lower interspecific variation in elevation and potentially greater environmental selection pressure. Therefore, we suggest prioritizing conservation efforts in areas with high species richness and focusing on species having small elevation ranges to develop sustainable and effective conservation strategies.

## Figures and Tables

**Figure 1 animals-14-01723-f001:**
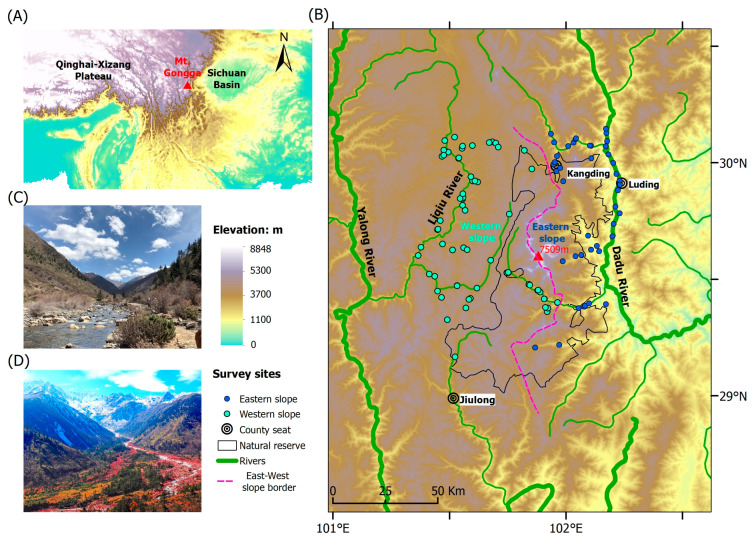
(**A**) Geographic location of the Gongga Mountains, (**B**) amphibian survey sites, and the typical habitats in the western (**C**) and eastern (**D**) slopes. The pink dashed line signifies the east–west slope border. The green and blue dots represent the survey sites on different slopes.

**Figure 2 animals-14-01723-f002:**
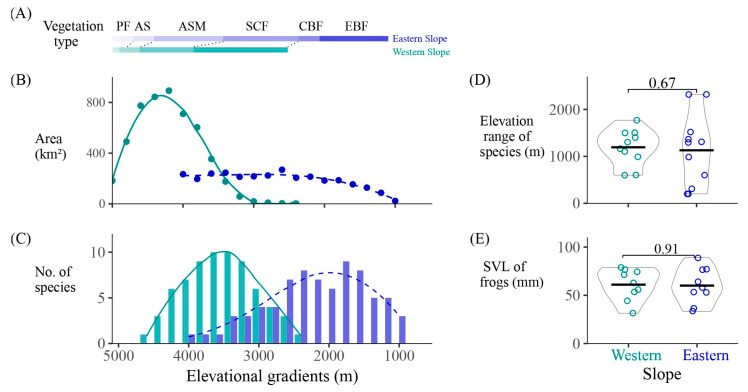
(**A**) The types of vegetation and (**B**) area, and (**C**) the number of amphibian species at different elevational gradients. (**D**) Elevation range and (**E**) snout-to-vent length of species on the western and eastern slopes. EBF, evergreen broad-leaf forest; CBF, coniferous and broad-leaf forests; SCF, subalpine coniferous forest; ASM, alpine scrub meadow; AS, alpine screes; PF, permafrost.

**Table 1 animals-14-01723-t001:** Amphibian species list and their local and overall elevation range and snout-to-vent length (SVL) on the eastern (E) and western (W) slopes of the Gongga Mountains.

Order	Family	Species	Slope of Mt. Gongga	Local Elevation Range: m	Overall Elevation Range: m	SVL: mm
Caudata	Hynobiidae	*Batrachuperus karlschmidti*	W	2997~4300	1800~4000	186
	Hynobiidae	*B. pinchonii*	E	2430~3950	1500~3950	155
	Salamandridae	*Liangshantriton taliangensis*	E	2100~2410	1390~3232	203
Anura	Bufonidae	*Bufo gargarizans*	W and E	1100~3764	120~4300	77
	Bufonidae	*B. tibetanus*	W	2702~4206	2400~4300	63
	Dicroglossidae	*Nanorana pleskei*	W	3100~4600	3300~4500	32
	Dicroglossidae	*Quasipaa boulengeri*	E	1500~1700	300~1900	89
	Hylidae	*Hyla annectans*	E	1150~1750	750~2400	34
	Megophryidae	*Megophrys shapingensis*	E	1668~2965	2000~3200	77
	Megophryidae	*M. minor*	E	1700~2686	300~2850	37
	Megophryidae	*Oreolalax major*	E	1600~1800	1600~2000	64
	Megophryidae	*Scutiger boulengeri*	W	2900~4300	2700~5100	54
	Megophryidae	*S. glandulatus*	W	2450~4220	2200~4000	79
	Megophryidae	*S. jiulongensis*	W	3329~3932	3120~3750	75
	Megophryidae	*S. mammatus*	W	3116~4220	2600~4200	72
	Ranidae	*Amolops loloensis*	E	1150~3469	2100~3200	58
	Ranidae	*A. mantzorum*	E	1283~2650	1000~3800	53
	Ranidae	*A. xinduqiao*	W	2655~3646	3300~3500	44
	Ranidae	*Rana chaochiaoensis*	E	1290~2600	1150~3500	54
	Ranidae	*R. kukunoris*	W	3200~3800	2000~4400	56

## Data Availability

The data presented in this study are available in the article and Appendix A.

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
