# Peer review of "Two-Dimensional Amphibian Diversity along a 3500 m Elevational Gradient at the Eastern Edge of the Qinghai–Xizang Plateau"

_animals, 2024, doi:10.3390/ani14121723_

Round 1
Reviewer 1 Report
Comments and Suggestions for Authors
Overall, I found this to be a sound study that contributes significantly to the knowledge of amphibian diversity in the region of the Gongga Mountains in China. However, I have questions regarding the methodology and analysis of genetic diversity. They are:
1. What species were used, what was the number of individual sequences for each species, and what is the GenBank accession number for each sequence? All of this information should be in the Supplementary files, and they are not.
2. Did the authors account for different number of sequences between the two slopes? In other words, is there greater nucleotide diversity on the eastern slope while controlling for the number of sequences? If the authors did not control for the total number of sequences in their analysis and there were more eastern-slope sequences than western-slope sequences, that alone would account for the greater number of haplotypes on the eastern side. If that is the case, the sections dealing with Genetic Diversity should be deleted as the results do not add anything to the paper.
3. If the authors did control for sequence number, there are other explanations than simply selection for differential genetic diversity. For example, in lines 273-274, the authors rightly point out that moisture enables amphibians to thrive. This means that amphibian populations in drier areas exist in smaller populations, which are more susceptible to genetic drift, which in turn lowers genetic diversity because of the random deletion of alleles in small populations (e.g., Marshall & Camp 2006. Environmental correlates of species and genetic richness in lungless salamanders (family plethodontidae). Acta Oecologica 299:33-44). Therefore, any greater genetic diversity on the eastern slope may simply be due to drift on the western slope, not selection.
4. Were there enough sequences to relate genetic diversity to elevation? This would be of interest if the data are available, the expectation being that genetic richness parallels species richness and peaks at mid elevations.
Line 270: “Ectotherm” is a better term than “poikilotherm” because amphibians do have the capacity for behavioral thermoregulation. “Poikilotherm” implies that they are simply thermal conformers to their environment, which they are not.
Line 291: I would avoid introducing the idea of competition. It introduces complexity that detracts from the overall story of the paper. Just because they are amphibians does not mean that the different species are competitors. The species present may have very different niche requirements and are not competitive at all. There simply may be fewer species because the warmer, drier environment does not enable as many species to exist at population levels that can sustain themselves.
Line 296: Explain the problem with fish farms. Is it because of habitat destruction to build the ponds, or do the presence of fish in the ponds result on predation pressure of amphibians trying to use them as breeding sites?
Comments on the Quality of English Language
One error that recurs is switching between present and past tense. Change all verbs related to the study to past tense (e.g., “we found that…). Use present tense when referring to a continuing pattern (e.g., “species A is listed as …” or “the mountains have an elevational range of…”).
Specific suggested changes with line numbers indicated:
31 Change to “indicators.”
37 Delete “that.”
40 Insert comma after “(western slope).”
42,43 Change to “But the species richness was unimodal, peaking at mid altitude … even though the greatest species richness occurred at different altitudes…”
44 Change to “types.”
55 Insert “global” before “decline,” and delete everything after “elevation.”
57,58 The difference between “low plateau with a mid-elevational peak” and “mid-elevation peak” is unclear. Explain.
71 Insert period after [14], and begin a new sentence with “They are good indicators…”
77 Delete “and amphibians” at the end of the sentence, and place “amphibians and” after “between.” Otherwise, it seems that amphibians is listed as an abiotic variable.
86 Delete comma.
95 Insert “with” after “faced.”
103 Change to “amphibians.”
117 Change “ranging” to “ranges.”
129,130 Delete “As for the western slope.”
132 Change “upward” to “higher.”
133 Change comma after “slope” to semicolon.
153 Change “inspected” to “inspecting.”
159,160 Change to “We collapsed observations of specimens from…one survey, resulting in…”
170 Delete “range.” Change “inquired” to “extracted.”
175 Change to “Amphibians were distributed across.”
199 Change “Referred to” to “In.”
210 Change “Above bars” to “Upper bars” or “Bars above.”
210-212 Change “means” equal sign (=).
215,216 Change to “…species) the other five families were distributed on both slopes but with different species replacing one another between slopes…”
219 Change to “Bufo gargarizans is distributed…”
222 Insert “in our study” after “smaller elevational range.”
231 Change “largely” to “which is.”
303-305 Change to” …abundant vegetation types. However, coniferous and broad-leaf forests (CBF) cover an elevation of only 300 m, which…”
319 Change “specie” to “species,” and delete “species” near end of line.
323 Change to “…species [51]; therefore, the…”
324 Change “great” to “greater.”
332 Delete “species.” Delete “the” before “people.”
Author Response
Comments and Suggestions for Authors
Overall, I found this to be a sound study that contributes significantly to the knowledge of amphibian diversity in the region of the Gongga Mountains in China. However, I have questions regarding the methodology and analysis of genetic diversity. They are:
- What species were used, what was the number of individual sequences for each species, and what is the GenBank accession number for each sequence? All of this information should be in the Supplementary files, and they are not.
- Did the authors account for different number of sequences between the two slopes? In other words, is there greater nucleotide diversity on the eastern slope while controlling for the number of sequences? If the authors did not control for the total number of sequences in their analysis and there were more eastern-slope sequences than western-slope sequences, that alone would account for the greater number of haplotypes on the eastern side. If that is the case, the sections dealing with Genetic Diversity should be deleted as the results do not add anything to the paper.
- If the authors did control for sequence number, there are other explanations than simply selection for differential genetic diversity. For example, in lines 273-274, the authors rightly point out that moisture enables amphibians to thrive. This means that amphibian populations in drier areas exist in smaller populations, which are more susceptible to genetic drift, which in turn lowers genetic diversity because of the random deletion of alleles in small populations (e.g., Marshall & Camp 2006. Environmental correlates of species and genetic richness in lungless salamanders (family plethodontidae). Acta Oecologica 299:33-44). Therefore, any greater genetic diversity on the eastern slope may simply be due to drift on the western slope, not selection.
- Were there enough sequences to relate genetic diversity to elevation? This would be of interest if the data are available, the expectation being that genetic richness parallels species richness and peaks at mid elevations.
We genuinely appreciate your thorough feedback. In the previous version of the manuscript, we only extracted one COI sequence from NCBI for each species. Subsequently, we calculated the genetic diversity based on 10 COI sequences (representing 10 species) from the western slope and 11 COI sequences from the eastern slope. Additionally, we calculated the genetic diversity within each elevational belt. We acknowledge that this method may not be entirely rational.
While we are interested in exploring the relationship between genetic diversity and elevation, we currently lack the necessary data (sequences) to do so. Some other reviewers have suggested removing the section discussing genetic diversity, and we have therefore decided to omit this information.
Line 270: “Ectotherm” is a better term than “poikilotherm” because amphibians do have the capacity for behavioral thermoregulation. “Poikilotherm” implies that they are simply thermal conformers to their environment, which they are not.
Thank you for your suggestion. we like ectotherm too and have displaced poikilotherm in this ms, as in Line 79 and Line 262.
Line 291: I would avoid introducing the idea of competition. It introduces complexity that detracts from the overall story of the paper. Just because they are amphibians does not mean that the different species are competitors. The species present may have very different niche requirements and are not competitive at all. There simply may be fewer species because the warmer, drier environment does not enable as many species to exist at population levels that can sustain themselves.
Thank you! We agree with you and have deleted the original sentences about competition, as in Line 279 now.
Line 296: Explain the problem with fish farms. Is it because of habitat destruction to build the ponds, or do the presence of fish in the ponds result on predation pressure of amphibians trying to use them as breeding sites?
Thank you! Local villagers have introduced invasive rainbow trout into the lake. These trout have the potential to consume the eggs and larvae of salamanders, encroach upon their breeding sites, and potentially carry lethal pathogens, as mentioned in Line 296-297.
Reviewer 2 Report
Comments and Suggestions for Authors
The manuscript revised is of serious value due to the weekly studied geographic region and increasing importance of ecological research for the conservation issues. The ms is well written, based on the using of adequate methods and comparative results.
However it should be improved after more discussion of analysis of biodiversity on the western and eastern slopes with explanations how the authors understand the differences on the slopes and their absence. The data on the influences of insolation and humidity will be useful in this context. The same as the conclusion that "The higher nucleotide diversity of the amphibian community in the eastern slope indicates weaker environmental selection pressure on species (Figure 2C)". However, the elevation range of species in the western slope is higher, but not significantly different between slopes (Figure 2D). It looks that the ms requires several more illustrations which can show the biodiversity on the both slopes.
Comments on the Quality of English Languagenone
Author Response
The manuscript revised is of serious value due to the weekly studied geographic region and increasing importance of ecological research for the conservation issues. The ms is well written, based on the using of adequate methods and comparative results.
However, it should be improved after more discussion of analysis of biodiversity on the western and eastern slopes with explanations how the authors understand the differences on the slopes and their absence. The data on the influences of insolation and humidity will be useful in this context. The same as the conclusion that "The higher nucleotide diversity of the amphibian community in the eastern slope indicates weaker environmental selection pressure on species (Figure 2C)". However, the elevation range of species in the western slope is higher, but not significantly different between slopes (Figure 2D). It looks that the ms requires several more illustrations which can show the biodiversity on both slopes.
We genuinely appreciate your feedback. In the previous version of the manuscript, we extracted only one COI sequence from NCBI for each species. Then we calculated the genetic diversity based on all 10 COI sequences on the western slope and 11 COI sequences on the eastern slope, as well as the genetic diversity within each elevational belt. However, following suggestions from other reviewers, we have decided to remove the section discussing genetic diversity entirely.
Additionally, we have removed the "genetic diversity" information and instead directly display the area (in km2) of each elevation belt in Figure 2C. We believe this adjustment will enhance the illustration of biodiversity (number of species) on both slopes.
Reviewer 3 Report
Comments and Suggestions for Authors
The manuscript titled “Two-dimensional amphibian diversity along a 3500 m eleva-2 tional gradient in the eastern edge of Qinghai-Tibetan Plateau” is in a good presentation and contains a very relevant study. All the finds were minutely detailed and provides important data for the understanding of herpetofauna diversity, specifically, to the Chinese amphibian diversity. So, I accept the manuscript with minor reviews described below.
- I think the data could be explored comparing also the time, and looking to the preserved area and an area not preserved. It also could be compared to the diversity of another area of study to identify the importance of delimiting a conserved area.
- The Authors could improve the analysis and results about genetic diversity, as they have a good amount of data on hands. The haplotype diversity could also brings more details about the diversity of the amphibians, for example. The evolutionary tree, seems not necessary once it could be just highlighted on a published tree, so the tree is simply representative, and not shows any result in fact.
Line 187:
To the genetic diversity you should use the portion of 12S and 16S mitochondrial fragment, as this is a fragment widely used for understand the evolutionary history and the kinship relationship between the species, or other taxa used. But this ion, because COI is also widely used.
Author Response
The manuscript titled “Two-dimensional amphibian diversity along a 3500 m elevational gradient in the eastern edge of Qinghai-Tibetan Plateau” is in a good presentation and contains a very relevant study. All the finds were minutely detailed and provides important data for the understanding of herpetofauna diversity, specifically, to the Chinese amphibian diversity. So, I accept the manuscript with minor reviews described below.
Thank you so much!
I think the data could be explored comparing also the time and looking to the preserved area and an area not preserved. It also could be compared to the diversity of another area of study to identify the importance of delimiting a conserved area.
Very good question. Given that the 60 surveys were conducted by different individuals, during various seasons, and using different methods, there is a potential for biased results when comparing data across time periods.
Comparing the amphibian community inside and outside the nature reserve is indeed a valuable idea. However, our current dataset is insufficient, and further investigation is warranted.
The Authors could improve the analysis and results about genetic diversity, as they have a good amount of data on hands. The haplotype diversity could also bring more details about the diversity of the amphibians, for example. The evolutionary tree seems not necessary once it could be just highlighted on a published tree, so the tree is simply representative, and not shows any result in fact.
We sincerely appreciate your comments. At present, we do not possess genetic data for these 20 amphibian species. The genetic diversity analysis was conducted solely based on COI sequences retrieved from NCBI, with only one sequence available per species. Consequently, we have removed the genetic diversity analysis from the manuscript.
You are correct that the tree does not present factual information. However, the evolutionary tree generated by Timetree can illustrate the relationships between species and provide insights into their evolutionary history. Therefore, it may be beneficial to retain it. Thank you for your input!
Line 187:
To the genetic diversity you should use the portion of 12S and 16S mitochondrial fragment, as this is a fragment widely used for understanding the evolutionary history and the kinship relationship between the species, or other taxa used. But this ion, because COI is also widely used.
Thank you, we agree. As the above response, the part of genetic diversity is deleted from ms now.
Reviewer 4 Report
Comments and Suggestions for Authors
Dear authors,
the ms includes a lot of descriptive data on the amphibian species inhabiting a remote mountain range in China. This is surely interesting but I believe that you chose the wrong journal, ms would be better suited in DIVERSITY.
As for the current ms, I made a lot of annotation in th ms pdf (attached). The methods concerning the main part of the study, the species identification in field and museum, are not described in detail. Therefore, the reliability of species list remains unkown. Many of the species are claimed threatened because of some local factors but these do not seem to have been assessed quantitatively (not mentioned in M&M and results), but are discussed. Moreover, based on data retrieved from public data bases such as GenBank, a phylogenetic tree is inferred which is not the state-of-art and does not really add significant information. The source of information on genetic diversity remains obscure. All genetic issues should be removed from the ms, as they are not original work of the authors and do not contribute to the message of the ms.
I recommend a thorough revision of the ms, focussing on the essentials of the study and transfering to Diversity.

The English shoud be checked and revised by a native speaker. I made some effort in improving terminology, but there are too many issues that need improvement.
Author Response
Dear authors,
the ms includes a lot of descriptive data on the amphibian species inhabiting a remote mountain range in China. This is surely interesting, but I believe that you chose the wrong journal, ms would be better suited in DIVERSITY.
We really appreciate your suggestion! We aggregated data from 60 surveys conducted over the past 8 years to illustrate the amphibian community and the two-dimensional distribution of richness within a global biodiversity hotspot. We anticipate that this paper will garner significant interest among readers in the field of animals.
As for the current ms, I made a lot of annotation in th ms pdf (attached). The methods concerning the main part of the study, the species identification in field and museum, are not described in detail. Therefore, the reliability of species list remains unknown.
Line 148. Information on species identification is completely lacking. Such a section is mandatory because it is often difficult to identify specimens in field (specifically, if they are juveniles) and in museum collections (loss of color, etc).
Line 155. Explain in detail: did you collect specimens? how did you determine species? etc.
Line 192. Without any information on the species identification procedure, this is a statement without foundation.
We really appreciate your thorough revision! It helps us a lot!
Indeed, the method of species identification is most important. We added detailed information about the method as in Line 147-153 and Line 156-158:
“Encountered animals (adults or juveniles) were captured and immediately identified by morphology referring to a color atlas of Chinese amphibians [41]. The animals were then released. For those that were difficult to confidently identified, specimens were collected and preserved for further examination in the lab. Larvae, in their early stages, were challenging to distinguish from each other. To avoid misidentification, larvae were morphologically identified only after the hind legs developed.”
“Most specimens stored in the museum were collected and identified by renowned herpetologists, including Chengzhao Liu, Liang Fei, Guanfu Wu, and Ermi Zhao.”
Many of the species are claimed threatened because of some local factors but these do not seem to have been assessed quantitatively (not mentioned in M&M and results) but are discussed.
We identified certain species as threatened based on their listing as Vulnerable in either the IUCN Red List or China’s Biodiversity Red List (2021). However, we further elucidate specific threat factors affecting these species in the Gongga Mountains.
Moreover, based on data retrieved from public data bases such as GenBank, a phylogenetic tree is inferred which is not the state-of-art and does not really add significant information. The source of information on genetic diversity remains obscure. All genetic issues should be removed from the ms, as they are not original work of the authors and do not contribute to the message of the ms.
Line 155. I do not see the rationale for this analysis. Please explain or delete.
Line 207. This not original work by you. As it is, it does provide deeper information to your study on geographic distribution.
We sincerely appreciate your comments. We currently lack genetic data for these 20 amphibian species. Genetic diversity has been assessed using COI sequences retrieved from NCBI, with only one sequence available per species. Consequently, we concur with removing the genetic diversity analysis from the manuscript.
Regarding the tree, you're correct in noting its lack of factual representation. However, the evolutionary tree generated by Timetree could illustrate species relationships and provide insights into their evolutionary history. Would it be possible to retain it? Thank you!
I recommend a thorough revision of the ms, focusing on the essentials of the study and transferring to Diversity.
Ulrich Sinsch
Line 163. How did you locate pertinent references?
We search reference using the species name as keywords.
Line 210. I do not understand this figure apart from the information on species richness at distinct altitudes. How did you obtain data on nucleotide diversity at distinct altitudes?
This figure is too complex. Better split it in few smaller figures given information on what you really did in your study, i.e., without data retrieved from other studies.
Thank you for your suggestion. We removed the nucleotide diversity and changed Figure 2C.
Line 213. This diagram does not make sense. why should one pool data from the body length of distinct anuran and urodele species???
Thank you! We modified the image and just compared the body length of frogs on the western and eastern slopes.
Line 216. All this information is better presented as a table, see upper comment.
Thank you. We added Table 1 in the ms.
Line 282. Reference
Thank you! I added one reference [51], as in Line 272.
Line 294. “Field surveys revealed that the main factors threatening amphibian species diversity in the Gongga Mountains are habitat degradation or loss, capture for food, and the presence of fish farms.”
This is new and a result out of place. If you wish to discuss this issue, you have to present hard data in the results section.
Thank you! We agree it should not be presented in the discussion and remove it.
Round 2
Reviewer 4 Report
Comments and Suggestions for Authors
The revised version of the ms is considerably improved compared with original one. The remaining issues are partly a matter of style.
1) Figure 2: I still believe it is too complex and should be split into two or three smaller ones.
2) The focus of the ms is on diversity, not so much on general issues. Therefore, you should consider Diversity as better suited than Animals.
Comments on the Quality of English Language
okay
Author Response
The revised version of the ms is considerably improved compared with original one. The remaining issues are partly a matter of style.
Thank you for your previous suggestions; they have greatly helped us improve the quality of our manuscript.
- Figure 2: I still believe it is too complex and should be split into two or three smaller ones.
We sincerely appreciate your comments. We've transferred some of the data from Figure 2 to Table 1. Additionally, we've revised Figure 2 to simplify it compared to its previous complexity.
2) The focus of the ms is on diversity, not so much on general issues. Therefore, you should consider Diversity as better suited than Animals.
Thank you for your recommendation. We have conducted extensive field investigations, specifically addressing the distribution and conservation efforts concerning amphibians. We are interested in submitting our manuscript to Animals.